# Scinderin Is a Novel Oncogene for Its Correlates with Poor Prognosis, Immune Infiltrates and Matrix Metalloproteinase-2/9 (MMP2/9) in Glioma

**DOI:** 10.3390/brainsci12101415

**Published:** 2022-10-20

**Authors:** Yiwei Wang, Zhongyan Li, Lili Bai, Dongyong Zhang, Tianchi Zhang, Fu Ren

**Affiliations:** 1Department of Anatomy, College of Basic Medical Sciences, Shenyang Medical College, Shenyang 110034, China; 2Department of Pathology, College of Basic Medical Sciences, Shenyang Medical College, Shenyang 110034, China; 3Shenyang Key Laboratory of Human Phenomics (SYKL-HP), Shenyang 110034, China; 4Department of Critical Care Medicine, Fuxin Central Hospital, Fuxin 123000, China; 5Department of Neurosurgery, First Affiliated Hospital of China Medical University, Shenyang 110001, China; 6Department of Computer and Information Technology, University of Pennsylvania, Philadelphia, PA 15419, USA

**Keywords:** glioma, scinderin, MMP2, MMP9, immune infiltration, prognosis

## Abstract

Purpose: The effect of scinderin (SCIN) on cancer progression has been studied, but its role in glioma remains unknown. This study describes the value of SCIN for the diagnosis, prognosis, and treatment of glioma. Methods: The expression of SCIN was analyzed using the GEPIA, Oncomine, cBioPortal, and CGGA databases. GO/KEGG enrichment analysis of similar genes to SCIN were performed using the R software package, and the protein–protein interaction (PPI) network was analyzed by the STRING and GeneMANIA databases. The correlations of mRNA expression between SCIN and MMP2/9 were analyzed by TCGA glioma. Simultaneously, the TISIDB and TIMER databases were used to analyze the correlation between SCIN and immune infiltration. Finally, SCIN and MMP2/9 protein expression among different grades of glioma was performed and the results were obtained via immunohistochemistry and Western blot assays. We used the Kaplan–Meier method and Cox proportional hazards model to assess the impact of SCIN and MMP2/9 on glioma patients’ survival. The correlations between SCIN and MMP2/9 were analyzed by immunohistochemistry and Western blot assays. Results: SCIN was upregulated in glioma patients with a poor prognosis. The GO and KEGG enrichment analysis showed the functional relationship between SCIN and the immune cell activation and regulation. In addition, the expression of SCIN was related to MMP2/9 in glioma. The correlation analysis showed that SCIN expression was associated with tumor purity and immune infiltration. SCIN and MMP2/9 are negative prognostic factors resulting in worsening glioma patients’ survival. Conclusion: Our studies demonstrated that SCIN expression was associated with MMP2/9, immune infiltration, and a poor prognosis in glioma. SCIN may serve as a potential prognostic marker and an immune therapy target for glioma.

## 1. Introduction

Glioma is one of the most frequent central nervous system (CNS) malignancies in adults. It is classified into four grades (I, II, III, and IV) by the World Health Organization (WHO) [1], grades I and II are the low grades, and grades III and IV are the high grades. Surgical resection remains the gold standard treatment for patients with preserved brain function, followed by radiotherapy and chemotherapy for high-grade gliomas. The mean survival time of patients with a high-grade glioma is only about 13 months, and that for a low-grade glioma is approximately 60 months [2]. Owing to the lack of molecular targets and effective therapeutic methods, the outcomes of the current therapy for high-grade gliomas remain unsatisfactory. Therefore, there is an increasing demand to find reliable biomarkers for diseases progression.

Scinderin (SCIN), or adseverin, is a calcium-dependent filamentous actin serving and capping protein that belongs to the gesoline superfamily [3,4]. SCIN is expressed in endocrine tissues and secretory cells. Recent studies suggest SCIN may be involved in the regulation of tumor development. For examples, Liu et al. found that SCIN played an important role in lung carcinoma cell proliferation [5]. Wang et al. showed that SCIN was highly expressed in human prostate cancer specimens and was associated with prostate cancer cell growth [6]. Liu et al. found high levels of SCIN expression in gastric cancer and promoted its invasion and metastasis [7]. However, Zhou et al. showed that SCIN was downregulated in hepatocellular carcinoma tissues and suppressed cell proliferation [8]. In brief, SCIN was suggested as a potential prognostic marker and a therapeutic target for some tumors, but the role of SCIN in tumor development and progression are controversial among different research projects. Currently, the expression and prognostic significance of SCIN, and its correlation with immune infiltration in gliomas, have not yet been explored.

The purpose of this study was to identify and evaluate SCIN as a molecular biomarker that can predict the risk, prognosis, and therapeutic responses. In this study, we found that SCIN was upregulated in gliomas, and a high expression of SCIN was associated with a poor survival. Furthermore, SCIN correlates with immune infiltration. In addition, the expressions of SCIN, MMP2, and MMP9 were associated with glioma prognosis. Clinicopathological analyses revealed that an elevated expression of SCIN in glioma patients was linked to an increased WHO grade and a poor survival, moreover, SCIN expression was positively correlated with MMP2 and MMP9 expressions.

## 2. Materials and Methods

### 2.1. Patients and Clinical Samples

This study was permitted by the medical ethics committee of the First Affiliated Hospital of China Medical University, and written informed consent was obtained from all patients. A total of 106 glioma tissue samples were collected from 2012 to 2018, and 10 non-neoplastic brain tissues specimens were collected from the available normal brain tissues post decompression surgeries for traumatic brain injury. None of the patients had received chemotherapy or radiotherapy before the surgeries. The glioma tissue samples were pathologically characterized by pathologists into grade I–IV according to the WHO glioma classification [1]. For additional details related to the patient materials, see Table 1. A tissue microarray was constructed from theses sample. Thirty-seven glioma samples were obtained from fresh surgical tissues frozen in liquid nitrogen for Western blot analysis.

### 2.2. Gene Expression Profiling Interactive Analysis (GEPIA)

The gene expressions were conducted by GEPIA (http://gepia2.cancer-pku.cn/#analysis, accessed on 3 January 2022) to compare the mRNA expression of SCIN between normal (GTEx samples) and tumor (TCGA samples) tissues [9]. Then, the relations of the disease-free survival (DFS) and overall survival (OS) rates with the expression of SCIN were computed by using the TCGA (LGG and GBM) database of GEPIA. SCIN-related genes in gliomas (LGG and GBM) were obtained from GEPIA and the top 200 similar genes are listed in Appendix A.

### 2.3. Oncomine and cBioPortal

The Oncomine database (http://www.oncomine.org/, accessed on 7 December 2021) was used to examine the mRNA level of SCIN in various types of cancers [10]. In this study, the t-test statistic was used to compare the control (normal brain tissue) and cancer specimens using the Oncomine database to generate a *p* value [11]. The *p* value and fold change were set at 0.05 and 2, respectively.

The cBioPortal (http://www.cbioportal.org/, accessed on 12 January 2022) was applied to investigate the genetic alternation of SCIN in gliomas. The UCSF dataset (LGG, UCSF Science, 2014, n = 61), TGCA dataset (GBM, TCGA Firehose Legacy, n = 604), MSKCC dataset (glioma, MSKCC Clin Cancer Res 2019, n = 1004), TGCA dataset (LGG, TCGA Firehose Legacy, n = 530), and TGCA dataset (merged cohort of lower-grade glioma and glioblastoma, TCGA Cell 2016, n = 1102) were analyzed in the cBioPortal.

According to the cBioPortal’s online instructions, the overall survival (OS) and disease-free survival (DFS) were analyzed to detect a with or without SCIN alternation in glioma [12].

### 2.4. Chinese Glioma Genome Atlas (CGGA)

We analyzed the mRNA SCIN expression in glioma of WHO grade II, III and IV, besides OS and DFS for high or low SCIN expression by three different datasets (mRNAseq_693, mRNAseq_325 and mRNA_array_301) in CGGA (http://www.cgga.org.cn/, accessed on 12 January 2022) [13].

### 2.5. Gene ontology (GO) and Kyoto Encyclopedia of Genes and Genomes (KEGG)

The GO enrichment and KEGG pathway analyses were used to assess the top 200 SCIN-related genes obtained from the LGG and GBM data on the GEPIA website (Appendix A). The results were performed by the ClusterProfiler package and by the ggplot2 package [14,15].

### 2.6. Protein–Protein Interaction (PPI) Network and GeneMANIA

The PPI network is composed of proteins and proteins through the interaction between each other. PPI was performed by STRING, an online database for the retrieval of interacting genes [16] and visualized using Cytoscape software (v3.7.1) [17].

GeneMANIA version 3.3.1 (http://genemania.org/, accessed on 25 January 2022) was used to perform gene groups with similar SCIN functions and to construct a network [16].

### 2.7. Immune Infiltration Analysis

The correlation between SCIN expression and the infiltration of immune cells of glioma was measured in tumor immune estimation resource (TIMER, http://timer.cistrome.org/, accessed on 25 January 2022) [18].

Tumor immune system interaction database (TISIDB, http://cis.hku.hk/TISIDB/index.php/, accessed on 25 January 2022) is a website for tumor and immune system interaction [19]. In this study, the interactions between SCIN and the contents of the immune system (tumor-infiltrating lymphocytes (TILs), chemokine, chemokine receptors, immunoinhibitor, immunostimulatory, and MHC molecule) were assessed in TISIDB.

### 2.8. The Cancer Genome Atlas (TCGA)

We also downloaded the count sequencing data of the GBM and LGG dataset TCGA-GBMLGG (TCGA, https://portal.gdc.cancer.gov/, accessed on 19 March 2022) [20] through the TCGA biolinks package [21]. They are normalized to FPKM (Fragments Per Kilobase per Million) forma. The dataset TCGA-GBMLGG has a total of 689 glioma samples. In this study, the correlations between SCIN and MMP2/9 were assessed by TCGA-GBMLGG and ggplot2 (version 3.0.0) for generating plots.

### 2.9. Tissue Microarray and Immunohistochemistry (IHC)

Glioma and normal brain tissue specimens were embedded in paraffin after being fixed with 4% paraformaldehyde. The tissues were then dehydrated with alcohol (70–100%) and permeated with xylene. Next, the samples were processed in a tissue microarray by a Manual Tissue Microarrayer (Quick-ray, UniTMA, South Korea). The tissue microarray sections were cut at 4 μm. The primary antibodies were Rabbit monoclonal anti-SCIN 1:100, (Abcam), Rabbit monoclonal anti-MMP-2 1:100 (Proteintech), and Rabbit monoclonal anti-MMP-9 1:100 (ABclonal). The UltraSensitiveTM SP (Mouse/Rabbit) IHC Kit (Maixin) was used as a secondary antibody kit. After being stained with DAB (3,3′-diaminobenzidine), the sections were observed with a microscope. For the control sections, the primary antibody was re placed by phosphate-buffered saline (PBS). The glioma cells immunoreactivity was scored according to their intensity and extent of staining. The intensity was scored as follows: 0 (negative staining), 1 (weakly positive), 2 (moderately positive), and 3 (strongly positive). The extent of positive stained cells was scored as follows: 0 (negative), 1 (1–25% stained cells), 2 (26–50% stained cells), and 3 (>50 positive cells). A multiplication of the intensity and extent of the positive stained cells scores was performed to calculate the immunoreactive scores and 4 was found to be the optimal cutoff value. The samples were classified into two groups: a low expression group (<4) and a high expression group (≥4).

### 2.10. Western Blot Assay

The total protein was extracted from glioma tissues using an RIPA buffer with a protease inhibitor. The protein concentrations were examined via the Bradford method using BSA as a standard. Then, the protein was resolved by SDS–PAGE electrophoresis and transferred to PVDF membranes, followed by 1 h 5% skimmed milk blocking at room temperature and then it was incubated overnight at 4 °C with a primary antibody (SCIN 1:1000, Abcam; MMP-2 1:1000, Proteintech; MMP-9 1:1000, ABclonal; GAPDH 1:5000, Abcam). After washing the membranes with TBST, the membranes were incubated with horseradish peroxidase-conjugated secondary antibodies anti-rabbit IgG and anti-mouse IgG (1:1000, Cell Signaling Technology) for 1 h at room temperature. The protein signals were developed by using an ECL reagent, and the densities were analyzed by Image Studio Lite software.

### 2.11. Statistical Analysis

The experimental results are showed as the means ± standard error of the mean. For continuous data, a *t*-test or ANOVA was used. The categorical data were measured using Fisher’s exact test or chi-squared analysis. The Kaplan–Meier method was compared using the log-rank test to assess differences in survival rates between the groups. A Cox proportional hazards model was used for univariate and multivariate analysis. Gene ontology (GO) enrichment and the Kyoto Encyclopedia of Genes and Genomes (KEGG) pathway analyses of co-expression genes were performed with the R (V 4.1) and R package “ClusterProfiler” for this analysis. Spearman correlation analysis was used to evaluate the relationship between SCIN and MMP2/MMP9. All the statistical analyses were conducted using GraphPad Prism 8.4 (GraphPad Software, La Jolla, CA, USA). Statistical significance was designated as follows: * *p* value < 0.05, ** *p* value < 0.01.

## 3. Results

### 3.1. SCIN Is Significantly Upregulated in Glioma

First, to identify SCIN expression in human cancers, we carried out GEPIA. We used different transcriptional expressions of SCIN in tumor and normal tissues using GEPIA and we found that SCIN expression was significantly differentially expressed in some cancers, especially in the LGG and GBM (Figure 1A). Then, the expression of SCIN in glioma and normal brain tissues was examined in the Oncomine database. Two brain and CNS datasets met the inclusion criteria, and the samples from the Sun Brain datasets using the Oncomine database corroborated this mRNA expression data (Figure 1B). Compared with that in normal brain tissues, the SCIN mRNA expression was higher in various glioma types, including diffuse astrocytoma, anaplastic astrocytoma, GBM, and oligodendroglioma. The frequencies of genetic alternations of SCIN in gliomas were evaluated using the cBioPortal database. The results showed that SCIN was altered in 1% (22/2225, altered/profiled) of 3156 patients/3301 samples in the UCSF dataset (LGG, UCSF Science, 2014, n = 61), TGCA dataset (GBM, TCGA Firehose Legacy, n = 604), MSKCC dataset (glioma, MSKCC Clin Cancer Res 2019, n = 1004), TGCA dataset (LGG, TCGA Firehose Legacy, n = 530), and TGCA dataset (merged cohort of lower grade glioma and glioblastoma, TCGA Cell 2016, n = 1102). These findings revealed that missense mutation or the amplification of SCIN occurs at a low rate in gliomas (Figure 1C). Then, we queried the expression of SCIN through GEPIA and found that SCIN expression was elevated in the LGG and GBM compared to normal brain tissues (Figure 1D). Further in the CGGA database, the SCIN high expression was associated with a high-grade glioma in three different datasets (mRNAseq_325, mRNAseq_693 and mRNA_array_301) (Figure 1E).

### 3.2. SCIN Is A Prognostic Factor for Glioma Patients

Given the association between the high SCIN expression and glioma grade, we speculated that SCIN could be a negative prognostic biomarker for the glioma outcome. To test our hypothesis, we analyzed the relationship between SCIN expression and the survival of glioma patients. We found that patients with high SCIN levels had a poor OS and DFS compared to those with a low SCIN level, using the GEPIA website (Figure 2A,B). Through the cBioPortal database, the correlation between the alteration in SCIN expression and survival of the cases was examined. As showed in Figure 2C,D, glioma cases with altered SCIN expression exhibited a significantly worse OS and DFS compared to those with unaltered SCIN expression. SCIN expression was associated with OS (Figure 2E), but not DFS (Figure 2F) in LGG. There were no statistical significance in DFS or OS between high and low SCIN expression groups in the GBM (Figure 2G,H). Moreover, CGGA analysis showed that SCIN expression was negatively associated with survival in glioma patients in three CGGA microarray datasets (Figure 2I).

### 3.3. Functional Enrichment Analysis of SCIN-Related Genes in Glioma

We used GO and KEGG analyses of the top 200 SCIN-related genes obtained from the LGG and GBM data in the GEPIA website (Appendix A). The SCIN-related genes were enriched in GO terms and the KEGG pathway related to immune cell proliferation, immune cell activation, immune cell adhesion, and cytokine biding (Figure 3A,B). Next, we performed the protein–protein interaction (PPI) network using the STRING and GeneMANIA databases (functional protein association networks). The top 10 hub proteins were retrieved by the STRING database: C3, CDH3, GPR87, GSN, LSP1, P2PY10, PIKFYVE, SIRT2, TMOD1, and TOMD2 (Figure 3C). Gene–gene interaction analysis using GeneMANIA showed 20 correlated genes with SCIN, such as CAPG, GSN, PIKFYVE, and TOMD2 (Figure 3D). We next utilized the TCGA datasets to conduct a correlation analysis between SCIN and MMP2/9. The correlation analysis observed that SCIN positively correlated with the expression of MMP2 (Spearman correlation coefficient r = 0.357, *p* < 0.001, Figure 3E) and MMP9 (Spearman correlation coefficient r = 0.356, *p* < 0.001, Figure 3F) in TCGA RNA seq datasets for glioblastoma and low-grade glioma (TCGA GBMLGG).

### 3.4. SCIN Expression Is Associated Immune Infiltration in Glioma Cells

Functional enrichment analysis showed that the SCIN-related genes were associated with the immune cell biological function. Therefore, we investigated a comprehensive analysis to reveal the correlation between SCIN expression and immune infiltration in gliomas based on the TISIDB and TIMER databases. In the TISIDB database, heatmaps showed the correlations between SCIN expression and tumor-infiltrating lymphocytes (TILs) (Figure 4A), chemokines (Figure 4B), chemokine receptors (Figure 4C), immuneinhibitors (Figure 4D), immunostimulators (Figure 4E), and a major histocompatibility complex (MHC) molecule (Figure 4F) in pan cancer. A positive correlation between SCIN expression and immune-related molecules was found in the LGG and GBM. The TIMER analysis showed that SCIN had significantly positive associations with infiltrating levels of the B cell, CD8+ T cell, CD4+ T cell, macrophage, neutrophil, and dendritic cell in the LGG and GBM (Figure 4G). Especially, SCIN was strongly correlated with the B cell, CD4+ T cell, macrophage, neutrophil, and dendritic cell in LGG. We further performed the Kaplan–Meier curve using the TIMER database to investigate the differences in the survival rates between high and low expression levels of SCIN and immune cells. We found that B cell infiltration (*p* < 0.001), CD8+ T cell (*p* < 0.01), CD4+ T cell (*p* < 0.001), macrophage (*p* < 0.001), neutrophil (*p* < 0.001), and dendritic cell (*p* = 0.001) correlated with LGG prognosis (Figure 4H). In GBM, the results showed the dendritic cell (*p* = 0.002) correlated with prognosis (Figure 4H) significantly (Figure 4H). The significant infiltration with immune cells seemed like one of critical factors that SCIN holds to influence the outcome of a glioma.

### 3.5. Co-Overexpressions of SCIN and MMP2/9 Are Associated with A Poor Prognosis

To gain further insight into SCIN and MMP2/9 changes that arise in glioma progression, TMA was made through our clinical specimens in our study, including 106 glioma tissues and 10 normal brain tissues, and the detailed clinicopathological characteristic data of the glioma patients are shown in Table 1. The TAM sections were stained for SCIN, MMP2, and MMP9 by IHC, respectively. We observed that SCIN and MMP2/9 protein expressions were upregulated significantly in glioma samples compared with normal brain tissues and progressively increased with the glioma grade (Figure 5A). As shown in Table 1, SCIN expression was positively correlated with the increasing WHO grade (*p* = 0.006), survival state (*p* < 0.001), MMP2 expression (*p* = 0.003), and MMP9 expression (*p* = 0.017). The Kaplan–Meier survival analysis of our clinical samples showed that an overexpression of SCIN, MMP2, and MMP9 were associated with a poor prognosis of glioma patients, respectively (Figure 5B). The SCIN and MMP9 expression levels were significantly associated with the OS of LGG. However, the effect of the MMP2 expression on the OS of LGG demonstrated no difference. In addition, associations of high SCIN, MMP2, and MMP9 expressions with the OS in GBM were not statistically significant (Figure 5D).

We further performed Western blot analysis to examine the expressions of SCIN, MMP2, and MMP9 in 37 fresh frozen glioma samples and 6 normal tissues. The results verified that the protein expression of SCIN and MMP2/9 were upregulated significantly in glioma tissues compared with normal brain tissues and were higher in high-grade (Grade III–IV) gliomas than in a low-grade (Grade II) glioma. Spearman correlation analysis was used to evaluate the relationship between the protein expression of SCIN and MMP2/9 in the fresh frozen tissues (Figure 6A–C). These results showed that SCIN expression was positively correlated with MMP2 (Spearman correlation coefficient r = 0.6848, *p* < 0.001, Figure 6D) and MMP9 (Spearman correlation coefficient r = 0.6218, *p* < 0.001, Figure 6E), respectively.

We also analyzed the effect of SCIN and MMP2/9 on the prognosis of glioma in our clinical samples. Univariate Cox analysis revealed that the expressions of SCIN (HR = 2.727, 95% CI = 1.608–4.625, Concordance Index = 0.622, *p* < 0.001), MMP2 (HR = 2.156, 95% CI = 1.273–3.650, Concordance Index = 0.595, *p* = 0.004), MMP9 (HR = 2.467, 95% CI = 1.442–4.220, Concordance Index = 0.58, *p* = 0.001), the WHO grade, and tumor diameter were risk factors in our clinical samples (Table 2). Multivariate Cox analysis showed that SCIN, MMP9, and the WHO grade were independently associated with survival in the clinical samples (Table 2).

## 4. Discussion

A glioma is one of the most common malignant tumors in the CNS. Whereas low-grade glioma patients may be cured through surgical treatment, HGG have a dismal prognosis despite the use of multimodality therapies including surgery, radiotherapy, and chemotherapy. The early molecular mechanisms underlying the proliferation, migration and invasion, which are crucial for a high-grade glioma treatment, remain poorly understood. Therefore, a novel biomarker correlated with progression is required for treatment with glioma. In the present study, SCIN has been identified as a new prognostic biomarker and associated with MMP2/9 expression and tumor immune infiltration in gliomas, which indicates that SCIN may be used as a target for glioma treatment.

Although the correlation between the SCIN expression and cancer progression has been revealed for a long time, multiple studies showed the role of SCIN in tumor development was controversial among different investigators. A high SCIN expression correlated with liver metastasis and poor progress in colorectal cancer. [22]. In gastric cancer, a high level of SCIN expression was associated with a poor prognosis of patients, and enhanced the proliferation, migration, invasion, and metastasis of gastric cancer cells [7,23]. Another study found SCIN was highly expressed in prostate cancer and promoted prostate cancer cell proliferation by the EGFR and MEK/ERK singling pathway [6,24]. Furthermore, SCIN was increased in breast cancer and the knockdown of SCIN could inhibit breast cancer cell proliferation and induce apoptosis [25]. However, in other research projects, the SCIN expression was low and associated with poor progress in acute myeloid leukemia, hepatocellular carcinoma, and gastric cancer [8,26,27]. Taken together, the above results demonstrate that SCIN plays an important role in mediating cancer cell proliferation, migration, invasion, and metastasis process. Our research is the first to propose the expression of SCIN in glioma, and also supports the oncogenic role of SCIN.

In our present study, we analyzed the expression and prognosis of SCIN in a large cohort of glioma patients from TCGA, CGGA, and our hospital samples. We revealed the expression of SCIN was higher than normal brain tissues and associated with glioma grade. Then, we observed that the missense mutation or amplification of SCIN occurred at a low rate in gliomas by the cBioportal. These results suggested that SCIN may play an important regulatory role in glioma progression. We then examined the prognostic value of SCIN in gliomas by the GEPIA, cBioportal, CGGA, and our hospital samples. We found that a high expression of SCIN indicated a poor prognosis. These results revealed SCIN was highly expressed in gliomas and associated with a poor prognosis in glioma patients.

During the past decades, the immunotherapy of gliomas is used in adjuvant surgery, radiotherapy, and temozolomide therapy [28,29]. However, the majority of glioma patients do not benefit from immunotherapy. The heterogeneous nature of glioma undermines the efficacy of immunotherapy [30]. Hence, understanding the tumor microenvironment was important to improve the immunotherapy of brain tumors [31]. In our study, we investigated GO and KEGG pathway analyses to conclude that SCIN-related genes were in several biological processes, including immune cell proliferation, immune cell activation, immune cell adhesion, and cytokine biding. By searching SCIN from the TISIDB website, we revealed a systematic difference between SCIN expression and TILs, chemokines, chemokine receptors, immunoinhibitors, immunostimulators, and MHC in LGG and GBM. The change in these immune-related factors will contribute to the improvement of the immunotherapy of gliomas. As shown in the TIMER database analysis, the expression level of SCIN was associated with infiltrating levels of the B cell, CD8+ T cell, CD4+ T cell, macrophage, neutrophil, and dendritic cell in the LGG and GBM. All the immune cells correlate with a poor prognosis in the LGG and the dendritic cell significantly correlate with a prognosis in the high expression group in the GBM. Our results indicate that SCIN can assess the status of the immune microenvironment and predict the prognosis of glioma patients.

Then, we constructed a PPI and GeneMANIA network to identify the SCIN interacting proteins. We found that GSN, PIKFYVE, and TMOD2 were expressed in the two networks. Gelsolin (GSN) is a Ca2+ regulated actin filament, and it impacts on cancer apoptosis and inflammation [32]. GSN is a biomarker candidate for GBM and inhibits glioma proliferation and invasion [33,34]. PIKFYVE (phosphatidylinositol 3,5-bisphosphate, PI (3,5) P2) is low abundant and involved in the membrane. PIKFYVE is critical for the regulation of autophagy, the postsynaptic function, embryonic development, and nervous system defects are a prominent subtype resulting from the disruption of PIKFYVE [35,36]. Tropomodulin 2 (TMOD2) encodes a neuronal-specific member of the tropomodulin family of actin-binding proteins in the central nervous system [37], which caps the minus end of actin filaments, preventing both elongation and depolymerization [38].

SCIN is an important actin-binding protein. It has been reported that the expression of SCIN is associated with the depth of invasion and lymph node metastasis of human gastric cancer and is related to a poor prognosis [7]. Moreover, a SCIN knockout reduces the migration ability of gastric cancer cells through regulating epithelial mesenchymal transformation (EMT) [23]. Previous studies have suggested that MMPs, especially MMP2 and MMP9, have been extensively demonstrated to promote progression, invasiveness, and a poor prognosis in gliomas [39,40] and are important factors in various pathological conditions of glioma, including tissue remodeling, morphogenesis, and especially the EMT of cells [41]. To further investigate the potential cause of the association between SCIN expression and the EMT of a glioma, we identified SCIN and MMP2/9 in our clinical tissue samples by IHC and Western blot analysis. We found that SCIN and MMP2/9 were upregulated in the primary glioma, SCIN had an excellent correlation with MMP2 and MMP9, and these proteins were closely related to the prognosis of a glioma. Considering the important role of SCIN and MMP2/9 in the occurrence and development of a glioma, we speculate that the high expression of SCIN combined with MMP2/9 may enhance the invasiveness of gliomas and lead to a more malignant tumor progression.

Our study demonstrated that SCIN expression was high and correlated with MMP2/9, a poor prognosis, and immune cell infiltration in gliomas. We hope our research will motivate further investigations and contributions to exploring the new mechanism of glioma progression. These results may facilitate clinicians to better understand the occurrence, development, and physiological characteristics of tumors, and provide important data for glioma diagnosis and patient prognosis.

## 5. Conclusions

In this study, SCIN was significantly associated with a poor prognosis, tumor immune infiltrates, and MMP2/9 expressions in a glioma. Uncovering the role of SCIN in the progression of a glioma brings us a step closer to developing a therapeutic target. However, the lack of cellular functional validation and the underlying molecular mechanisms are potential limitations to our study. Future studies will aim to elucidate the molecular mechanism involved in mediating the functional expression of SCIN.

## Figures and Tables

**Figure 1 brainsci-12-01415-f001:**
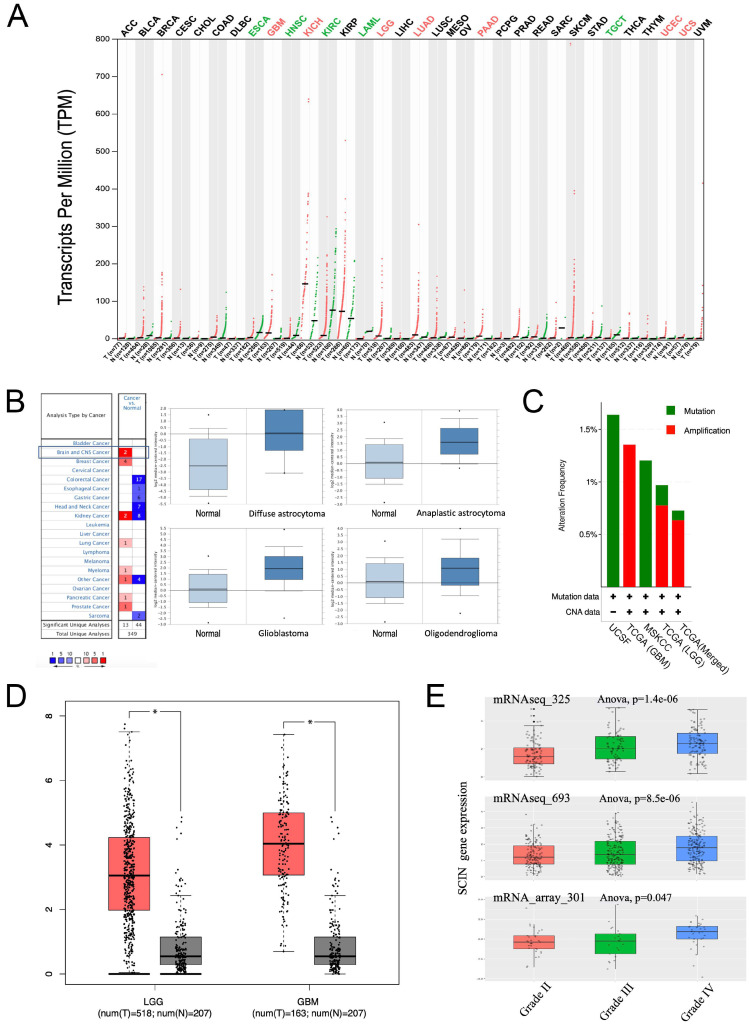
Scinderin (SCIN) expression is elevated in gliomas. (**A**) SCIN transcriptional expression levels in different cancers compared to normal tissues from The Cancer Genome Atlas (TCGA) and GTEx Portal databases analyzed by Gene Expression Profiling Interactive Analysis (GEPIA). There is a statistical difference (* *p* < 0.05). (**B**) SCIN expression level of different cancers in Oncomine Database. The left picture shows that there are two SCIN hyperexpression in the (central nervous system) CNS datasets after comparing cancerous and normal tissues. The right part is a similar analysis of samples from the Sun Brain dataset (* *p* < 0.05). Compared with that in normal brain tissues, the SCIN mRNA expression is higher in various glioma types, including diffuse astrocytoma (fold change = 4.522, t = 2.931, *p* = 0.006), anaplastic astrocytoma (fold change = 2.807, t = 3.815, *p* = 2.32 × 10^−4^), glioblastoma (fold change = 3.348, t = 5.181, *p* = 4.20 × 10^−6^), and oligodendroglioma (fold change = 1.727, t = 2.185, *p* = 0.017). (**C**) cBioPortal analysis of the SCIN alternation rate in glioma. (**D**) SCIN level in lower-grade glioma (LGG) and glioblastoma (GBM) compared to levels in normal brain tissues by GEPIA dataset. (**E**) SCIN expression in glioma of WHO grade II, III, and IV in Chinese Glioma Genome Atlas (CGGA) datasets. CNA, copy number alteration.

**Figure 2 brainsci-12-01415-f002:**
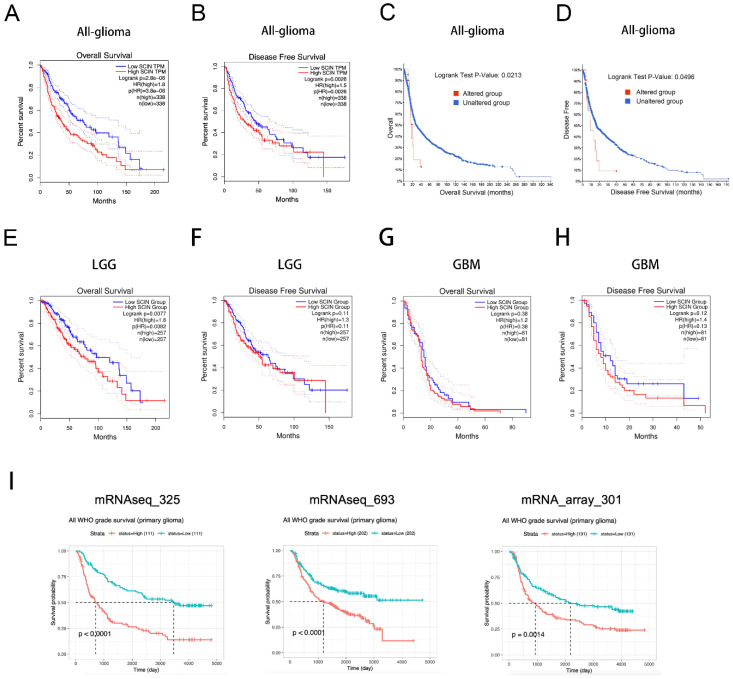
Scinderin (SCIN) is a prognostic biomarker in glioma. (**A**) Overall survival (OS) in glioma patients with high or low level of SCIN analyzed in the Gene Expression Profiling Interactive Analysis (GEPIA)dataset (*p* < 0.001). (**B**) Disease-free survival (DFS) in glioma patients with high or low level of SCIN analyzed in the GEPIA dataset (*p* = 0.0026). (**C**) OS in glioma patients was divided into two groups according to SCIN alternation (*p* = 0.0213). (**D**) DFS in glioma patients was divided into two groups according to SCIN alternation (*p* = 0.0496). (**E**,**F**) OS (*p* = 0.0077) and DFS (*p* = 0.11) analysis based on SCIN expression level in LGG from the GEPIA dataset. (**G**,**H**) OS (*p* = 0.38) and DFS (*p* = 0.12) in GBM from the GEPIA dataset. (**I**) Chinese Glioma Genome Atlas (CGGA) analysis of the prognostic significance of SCIN expression in glioma patients in three datasets (*p* < 0.01).

**Figure 3 brainsci-12-01415-f003:**
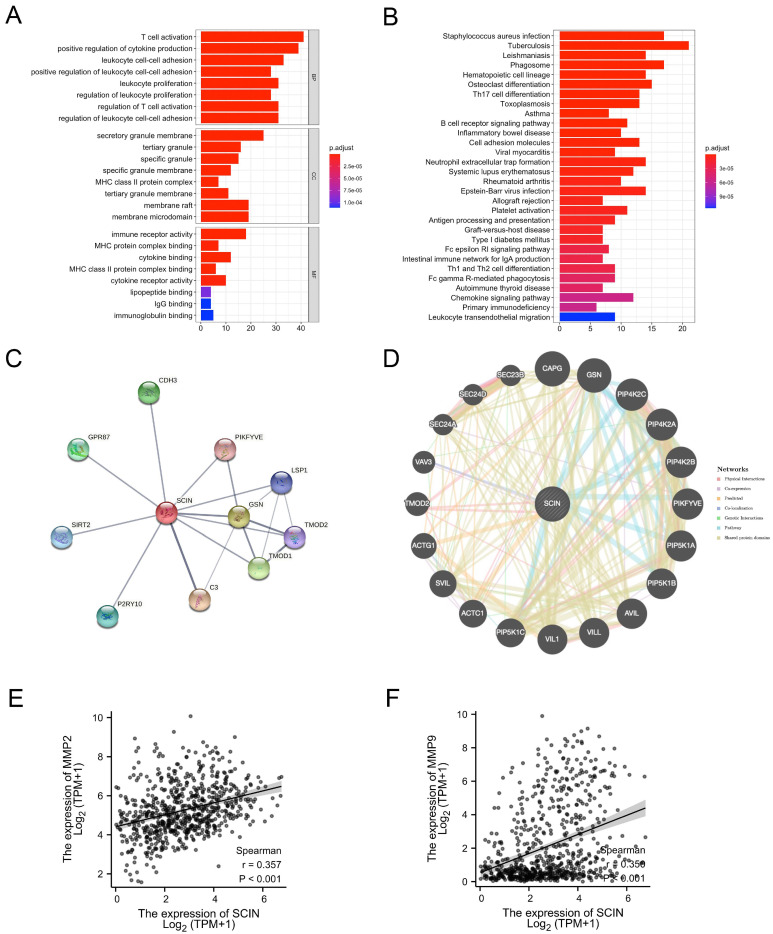
Interaction analysis of scinderin (SCIN). (**A**) Top 10 Gene ontology (GO) terms, involving biological process (BP), cellular component (CC), and molecular function (MF) in SCIN similar genes in glioma. (**B**) Top 30 Kyoto Encyclopedia of Genes and Genomes (KEGG) pathway in SCIN similar genes in glioma. (**C**) The Protein–Protein Interaction (PPI)network of 10 proteins constructed with STRING server. (**D**) Gene–gene interaction network for SCIN constructed by GeneMANIA prediction server. (**E**,**F**) Correlation analysis results of SCIN with MMP2 (r = 0.357, *p* < 0.01) and MMP9 (r = 0.356, *p* < 0.01) in the The Cancer Genome Atlas (TCGA) dataset.

**Figure 4 brainsci-12-01415-f004:**
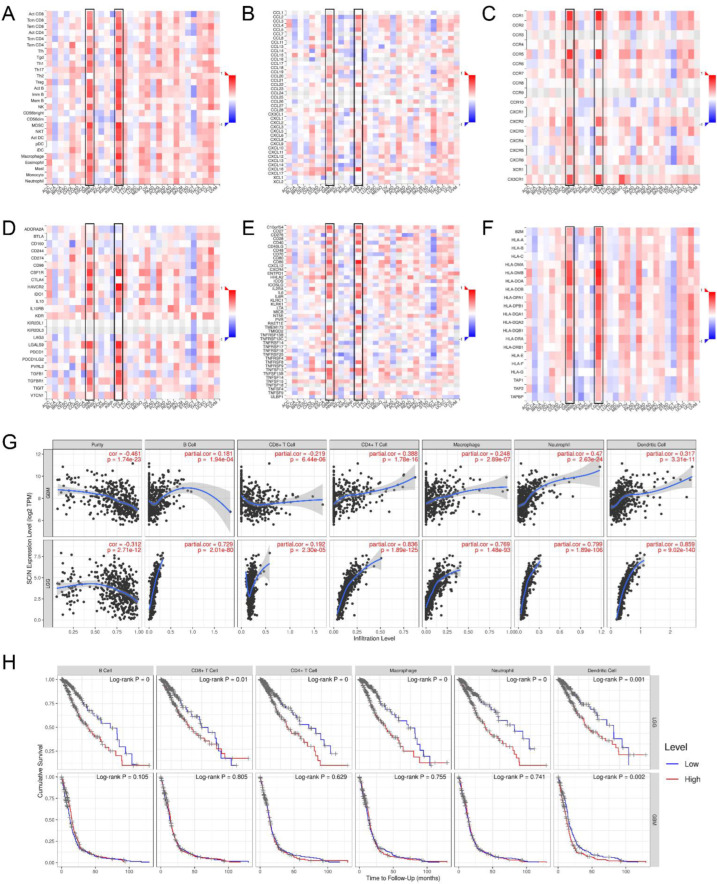
Associations of scinderin (SCIN) expression level with immune cell infiltration. (**A**) Correlation between SCIN and abundance of tumor-infiltrating lymphocytes (TILs) in tumor immune system interaction database (TISIDB). (**B**) Correlation of SCIN with chemokine in TISIDB. (**C**) Correlations between SCIN and chemokine receptors in TISIDB. (**D**–**F**) Correlations between SCIN expressions and immunomodulators (immunoinhibitor, immunostimulatory, and MHC molecule) in TISIDB. Red is positively correlated. Blue is negatively correlated. (**G**) Association between SCIN and immune infiltration in GBM and LGG. (**H**) Cumulative survival is related to B cell, CD8+ T cell, CD4+ T cell, macrophage, neutrophil, and dendritic cell in LGG and GBM.

**Figure 5 brainsci-12-01415-f005:**
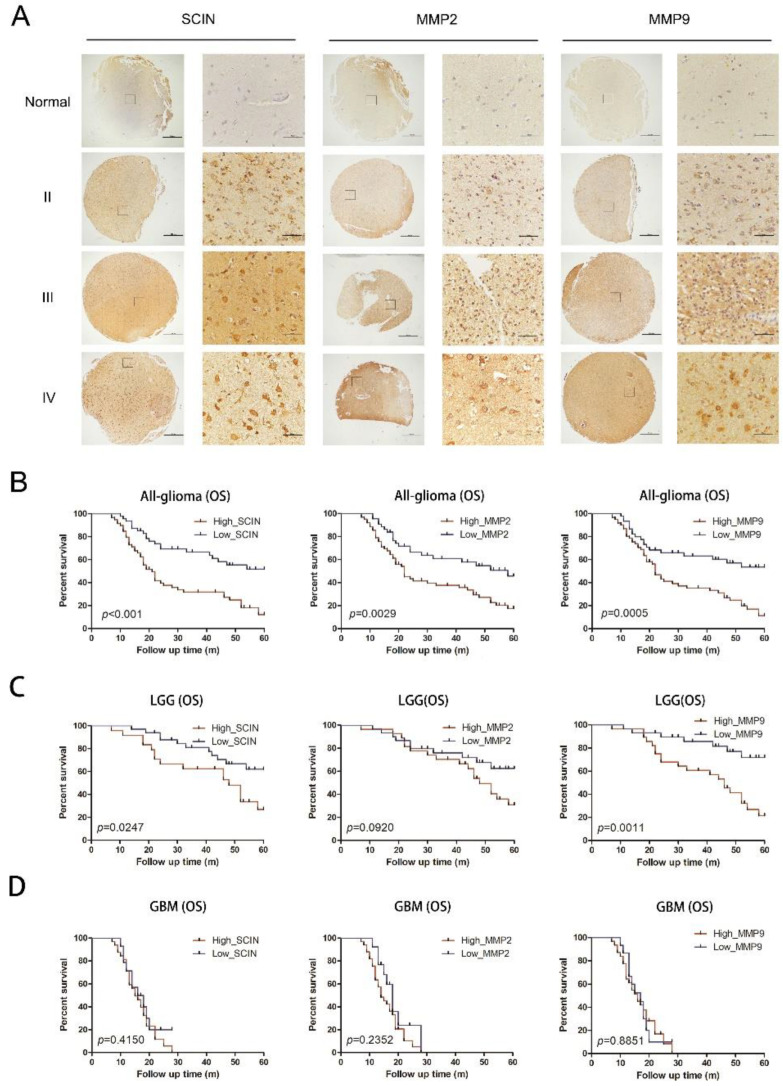
Expression of SCIN and MMP2/9 in glioma and Kaplan–Meier analysis. (**A**) The protein expression of SCIN, MMP2, and MMP9 in glioma and normal brain tissues through IHC (Scale of TAM: bar = 500 μm, zoom in section: bar = 100 μm). (**B**) The Kaplan–Meier OS curves in glioma patients provided significant differences between SCIN (*p* < 0.001), MMP2 (*p* = 0.0029) and MMP9 (*p* = 0.0005). (**C**) OS analysis in LGG based on the expressions of SCIN (*p* = 0.0274), MMP2 (*p* = 0.0920), and MMP9 (*p* = 0.0011) using Kaplan–Meier curves (**D**) OS analysis in GBM based on SCIN (*p* = 0.415), MMP2 (*p* = 0.2352), and MMP9 (*p* = 0.8851).

**Figure 6 brainsci-12-01415-f006:**
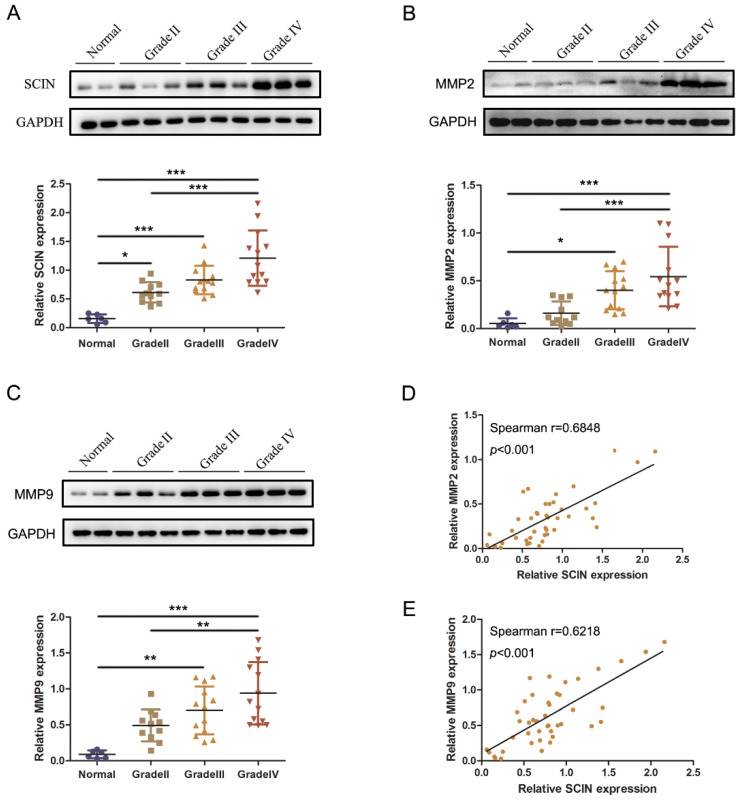
The expression of SCIN is correlated with MMP2/9. (**A**) Western blot images and analyses of SCIN expressed in 6 normal and 37 glioma tissues (* *p* < 0.05. *** *p* < 0.001). (**B**) Western blot images and analyses of MMP2 expressed (* *p* < 0.05. *** *p* < 0.001). (**C**) Western blot images and analyses of MMP9 expressed (** *p* < 0.01. *** *p* < 0.001). (**D**,**E**) Correlation analysis between SCIN and MMP2/9 in our clinical tissues (r = 0.6848, *p* < 0.001 and r = 0.6218, *p* < 0.001, respectively).

**Table 1 brainsci-12-01415-t001:** Clinical features and SCIN expression in 106 glioma patients.

Clinical Features		Cases	High SCIN Expression	Low SCIN Expression	*p* Value
Age (Mean ± SD, years)		106	52.86 ± 11.488	56.81 ± 9.518	0.380
Gender	Male	49	24	25	0.199
	Female	57	35	22	
Tumor size (cm) < 3		35	16	19	0.148
	≥3	71	43	28	
KPS	<80	62	38	24	0.166
	≥80	44	21	23	
WHO Grade	I	3	1	2	0.006 *
	II	31	10	21	
	III	26	15	11	
	IV	46	33	13	
Survival state	Alive	40	13	27	<0.001 *
	Death	66	46	20	
MMP2 expression	High	62	42	20	0.003 *
	Low	44	17	27	
MMP9 expression	High	61	40	21	0.017 *
	Low	45	19	26	

Notes: KPS, Karnofsky performance scale; WHO, World Health Organization. Statistical analyses were performed by the Fisher’s exact test. * *p* < 0.05 was considered statistically significant.

**Table 2 brainsci-12-01415-t002:** Univariate and multivariate analyses of various prognostic parameters in patients with gliomas.

	Univariate Analysis				Multivariate Analysis			
	*p* Value	Hazard Ratio	95% Confidence Interval	Concordance Index	*p* Value	Hazard Ratio	95% Confidence Interval	Concordance Index
SCIN	<0.001	2.727	1.608–4.625	0.622	0.034	1.857	1.047–3.292	
MMP2	0.004	2.156	1.273–3.650	0.595	0.695	0.887	0.488–1.613	
MMP9	0.001	2.467	1.442–4.220	0.588	0.039	1.851	1.032–3.317	0.723
Tumor size	0.024	1.914	1.088–3.366	0.567	0.297	1.368	0.760–2.463	
WHO Grade	<0.001	6.048	3.074–11.902	0.672	<0.001	5.300	2.499–11.240	

## Data Availability

The data used to support the findings of this study are included within the article.

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
