# Peer review of "Scinderin Is a Novel Oncogene for Its Correlates with Poor Prognosis, Immune Infiltrates and Matrix Metalloproteinase-2/9 (MMP2/9) in Glioma"

_brainsci, 2022, doi:10.3390/brainsci12101415_

Round 1

Reviewer 1 Report

In this study, Wang et al. evaluate the expression of SCIN in LGG and HGG, the correlation of this gene with immune infiltration, as well as the expression of SCIN, MMP2 and MMP9 correlating with glioma's prognosis.

The analysis was performed using several databases available online. The relevance of SCIN in gliomas can be considered a novelty, however, the use of different databases makes it difficult to follow the results and the conclusions of this work. Did the authors consider the use of a single database, or to extract the relevant data from multiple databases, combine them and perform similar analysis (using R platform, for example)?

The analysis of the overall survival (OS) and the disease free survival (DSF) was performed based on the data of all gliomas. It was shown a lower expression of SCIN by LGG, which presents better prognostic compared with HGG, which has poor prognosis. Accurate interpretation of the OS and DSF should be performed segregating the tumours by grade. 

The correlation analysis of SCIN with MMP2/9 showed a low R-squared value in the Cox Regression Model. Regardless of the variable significance, a low R-squared might indicate that the independent variable is not explaining much in the variation of the dependent variables. The interpretation of this data must be accurate. For low R-squared values, additional measures (e.g. Concordance Index) must be performed to evaluate the usefulness of the model and increase the confidence of the results obtained. 

The work is consistent with the aim of the research, but the English needs major corrections. 

I have several comments that need some action. 

line 48: "Owing to the lack of molecular heterogeneity..." - high grade gliomas are known to present a high intra and inter cellular heterogeneity. The authors might want to reformulate this sentence.

line 53: "...capping protein that belongs to the gasoline superfamily" - SCIN belongs to the gelsolin super family of actin binding proteins, not gasoline. The authors should correct this. 

line 61: "...potential diagnostic marker..." - Do the authors want to mention diagnostic marker or prognostic marker? Diagnostic marker is a biological parameter that helps the diagnosis of the disease. 

lines 70-71: "In addition, The expressions of SCIN, MMP2 and MMP9 were associated with glioma prognosis and their relatioships with clinicopathological characters were analyzed. .'' some misspelling words. As a follow up of the main findings, the relationship with the clinicopathological features should be described.

line 81: "Table 1" describes the top 200 SCIN related genes in LGG and GBM. 

line 89: a reference is missing for the Oncomine database.

lines 99-101: It is not clear that the TISIDB database was used and how the analysis was performed. 

line 105: "...dehydrated gradiantly and xylene was transparent." this is not clear. 

line 110: an extra paragraph was added. 

Line 123: Results from glioma cell lines were not included in this study. 

Line 125: Did the authors resolve the proteins by electrophoresis (e.g. 10 % SDS-PAGE).

Line 150: the content is not clear. 

Line 155: "...Sun Brain datasets was further analysis." - were further analyzed? This is not clear. 

Line 163: "These founding revealed that missense mutation or amplification of SCIN occurs at a low rate in glioma" - These findings? 

Line 165: "Then we queried the expression of SCIN through GEPIA and found that SCIN was significantly differentially expressed in LGG and GBM (Figure 1D)." - Compared to normal brain tissue? What about the comparison between LGG and GBM using this database?

Line 177: further analysed?  The description of figure 1 should be improved. 

Line 184: Section 3.2- The analysis of the OS and DFS was performed taking in consideration all gliomas. Since the LGG gliomas are showing lower expression of this gene, I am afraid that the results are influenced by the tumour grade. To have some significance, the same analysis should be performed segregating the tumour by grade. 

Line 213: "We observed a highly significant, correlation analysis between SCIN and MMP2..." - the wording should be more clear. 

Line 215: values of coefficient R between 0.2 and 0.4 are considered a weak correlation. 

Suggestion: Table 1 is huge, it should be added as a supplementary table. PPC must be described. 

Line 227: Section 3.4. The wording/syntax must be reviewed in this section. 

Line 256: Review all section 3.5. Not clear.

Line 302: "...incurable adjuvant radiotherapy and chemotherapy." is missing a verb.

The conclusion must be reviewed. Several misspelling and grammatical errors.

Lines 344-346: "As shown in TIMER database analysis also shows that the expression level of SCIN was associated with infiltrating levels of B cell, CD8+ T cell, CD4+ T cell, macrophage, neutrophil and dendritic cell in LGG and GBM." Is that true for GBM?

Line 386: "Our study demonstrates that SCIN is up-regulated in glioma and significantly associated with MMP2 /9 and glioma prognosis" - The weak correlation of SCIN with MMP2/9 must be considered in the conclusion. 

Reviewer 2 Report

Authors investigate the role of SCIN in glioma. The paper is well structured, still lacks details on several occasions, listed without prominence as appeared along the text:

  • line 66 purpose was to investigate the role of SCIN or other?

  • reference to "gasoline superfamily" missing?

  • method details and references missing to: cluster profiler version, GEPIA, Oncomine database, cBioPortal, CGGA, TimerDB, Genemania, StringDB: the particular information in regard to which data sets (cbioportal?), settings used, version or date of site accessed and check the How to cite details for each resource!

  • Please refine "additional details related to patient materials". Additionally, Table 1 legend should be clarified, data comes from clinical samples collected as you refer in the materials section (wrogn numbering as to table 2?) or databases used as stated in the results? PCC - pearson correlation coefficient?

  • Anonymized clinical sample data could be provided via a relevant repository and the data availability statement updated. It could also include information on data sets used from others.

  • Figure 1 A colours do not match expression levels as indicated by TPM, please refine description - in relation to normal equivalent tissue or else

  • The conclusion could be refined in terms of limitations of the study and future outlook also regarding next steps.

Round 2

Reviewer 1 Report

The authors had addressed the major issues of this work, increasing its quality. However, the following concerns should be taking in consideration.  

Line 223: "foundings"  replace by findings.

Line248: regards the "Response 2:

Thank you for your hard work, we completely agree with you. Our present study in patients with gliomas (GBM and LGG) showed that SCIN expression was associated with DFS and OS. However, the expression of SCIN had no significant effect on the OS of GBM and the DFS of GBM or LGG alone in GEPIA. We also found that this phenomenon is consistent with the research in our clinical samples, which is worthy of further study. We venture speculate that this indicator may be the highly overexpressions of SCIN in GBM(Figure 1), and it is difficult to effectively differentiate scores of prognosis. Therefore, we only show the results of DFS and OS in glioma. Thank you for your great common ."

As authors are confirming that the expression of SCIN had no significant effect on OS of GBM and DFS of GBM and LGG. I assume that it is only significant for OS of LGG. Presenting the results like they are leads to a misinterpretation of the real value of SCIN in terns of prognostic marker. Like they segregate the tumours by grade in Figures 1, 4, and 5, and table 2, the same should be done for figure 2, even if the study was performed in patient with gliomas. 

Line 370: The same should be done with the authors are showing the OS of SCIN, MMP2 and MMP9 in their clinical samples. MMP2 and MMP9 are overexpressed in GBM compared to LGG, as the authors showed in Figure 6.
